# High-Temperature Equal-Channel Angular Pressing of a T6-Al-Cu-Li-Mg-Ag-Zr-Sc Alloy

**Marcello Cabibbo** 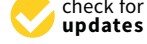 **and Chiara Paoletti \*** 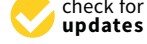

Department of Mechanical Engineering and Mathematical Sciences (DIISM), Università Politecnica delle Marche, Via Brecce Bianche 12, 60131 Ancona, Italy; m.cabibbo@staff.univpm.it
**\*** Correspondence: c.paoletti@pm.univpm.it

**Abstract:** Equal-channel angular pressing (ECAP) is known to induce significant grain refinement and formation of tangled dislocations within the grains. These are induced to evolve to form low-angle boundaries (i.e., cell boundaries) and eventually high-angle boundaries (i.e., grain boundaries). On the other hand, the precipitation sequence of age hardening aluminum alloys can be significantly affected by pre-straining and severe plastic deformation. Thus, ECAP is expected to influence the T6 response of aluminum alloys. In this study, a complex Al-Cu-Mg-Li-Ag-Zr-Sc alloy was subjected to ECAP following different straining paths. The alloy was ECAP at 460 K via route A, C, and by forward-backward route A (FB-route A) up to four passes. That is, ECAP was carried out imposing billet rotation between passes (route A), billet rotation by +90° between passes (route C), and billet rotation by +90° and inversion upside down between passes (FB-route A). The alloy was also aged at 460 K for different durations after ECAP. TEM microstructure inspections showed a marked influence of the different shearing deformations induced by ECAP on the alloy aging response. The precipitation kinetics of the different hardening secondary phases were affected by shearing deformation and tangled dislocations. In particular, the $T_1$-Al$_2$CuLi phase was the one that mostly showed a precipitation sequence speed up induced by the tangled dislocations formed during ECAP. The T1 phase was found to grow with aging time according to the Lifshitz-Slyozov-Wagner low-power regime.

**Keywords:** ECAP; Al-Cu-Mg-Li-Ag-Zr-Sc alloy; secondary-phase precipitates; tangled dislocations; alloy aging; transmission electron microscopy



## 1. Introduction

Al-Cu and Al-Cu-Mg alloys are widely used in the aircraft industry due to their excellent mechanical properties, which depend on the fine dispersion of secondary phase strengthening precipitates [1–4].

Al-Cu-Li based alloys are recognized as important metallic materials for structural applications requiring a combination of high strength, low density, high fracture toughness, and good corrosion resistance [3,5–8]. All these mechanical and physical characteristics are particularly relevant for aerospace applications [5–7]. Moreover, weight reduction is now widely considered as a primary means to lower fuel cost in the aeronautics and aerospace industry [7]. A reduction in aircraft weight reduces fuel consumption, thus increasing the payload capability [3]. Accordingly, lithium, as the lightest metal known, was used as an additional element in a number of aluminum series alloys. An addition of 1 wt.% Li reduces the Al density by 3%, and usually increases the alloy elastic modulus by some 6% [9].

Thus, starting from the first generations of Al-Li alloys, developed in the 1980s, the current alloy developments concern low Li contents balanced by addition of other elements such Mg, Mn, Zr, Ca, Ag, RE, and so forth. These alloying modifications greatly helped to improve some of the weak points of the Al-Li alloys, such as the limited long-term

stability [10,11]. In the last three decades, scientific research on Al-Li-based alloys mostly focused on Al-Cu-Li, Al-Cu-Mg, and Al-Mg-Li alloys [12,13].

In fact, the mechanical properties, together with most of the physical properties of Al-Cu-Li alloys are driven by the fine scale distribution of precipitate phases. These are favored to form by solutioning followed by aging at different temperatures and duration, depending on the specific alloying elements added to the Al-Cu-Li base alloy. In the base Al-Cu-Li system, the two secondary phase precipitates that are induced to form belong to the binary Al-Cu and Al-Li, and ternary Al-Cu-Li systems. Thus, the precipitating phases are GP-I, GP-II zones, $\theta''$, $\theta'$ phase, to end at the stable $\theta$ ($Al_2Cu$), and with a similar formation process of the $\acute{o}'$ to $\acute{o}$ ($Al_3Li$) and $T_1$ ($Al_2CuLi$) phases [14–19].

Addition of other alloying elements (especially Mg, Mn, Ag, Zn, Zr) results in a wider variety of precipitating secondary phases. As for the role of Ag, in the complex Al-Cu-Li-X alloy, it was widely documented that it tends to segregate in the $T_1$-$Al_2CuLi$ phase and in the $S'$-$Al_2CuMg$ phase, thus contributing to their microstructure strengthening effects [20].

Together with the above-mentioned strengthening fine-dispersed precipitation phases, the lithium-containing aluminum alloys are engineered in terms of ductility. In this respect, the refining processes of both grains and cells are important microstructure features for improving the alloy mechanical properties. Hence, alloying elements, heat treatments and plastic deformation are the three major means to tailoring a sound and technologically interesting Li-bearing complex aluminum alloy [21–23].

The AA2198 (Al-Cu-Li-Mg-Ag-Zr), and similar alloys, sometimes referred to as Weldalite® alloys [24], are known to be characterized by the formation of all the above-mentioned secondary phases that are induced to precipitate under annealing and peak ageing.

In particular, the $\theta$ ($Al_2Cu$) phase is tetragonal with lattice parameters $a$ = 4.04, and $c$ = 5.80 Å, and a disc-like or platelet morphology typically oriented along $Al_{100}$ crystallographic plane, which is similar to a [001]-precipitate plane ‖ $Al_{100}$ matrix [25,26]. The effective hardening compound is the GP-II/$\theta'$ phase, which is essentially a semi-coherent precipitate. This forms by vacancy releasing and GP-I agglomerate formation via Cu atoms short-range diffusion [27,28]. This is the alloy-strengthening phase for Al-Cu binary alloys.

On the other hand, by adding Li to the alloy, nucleation of the $T_1$ phase constitutes the highest nanometric scale alloy-strengthening feature. This phase has a hexagonal (hcp) structure with lattice parameters $a$ = 4.97, and $c$ = 9.35 Å, and plate-like morphology with large aspect ratio of which the longer edge (needle) lying on $[111]_{Al}$ planes and orientation relationship $(0001)$-$T_1$ ‖ $(100)_{Al}$. In particular, this phase shows its maximum strengthening effect when the needle-shaped edge of the precipitates lye as to have $(0001)_{T_1}$ habit planes ‖ $(111)_{Al}$, and a $[10,11]_{T_1}$ ‖ $Al_{110}$ matrix plane. The $T_1$ phase has a space group $P6/mmm$ [29] and it has been shown to be shearable by dislocations introduced during plastic deformation techniques [29–31]. In particular, the precipitate shearable planes are as to have $[10,11]_{T_1}$ ‖ $[110]_{Al}$ planes, i.e., a $[10,11]_{T_1}$ crystallographic direction. When Mg and Ag are added to the Al-Li based alloys, these are known to easily cluster within grain and cell interiors eventually accelerating the nucleation of the $T_1$ phase upon aging [29–31].

Moreover, a number of research findings showed that the generation of the hardening $T_1$ phase is promoted by the addition of minor solute elements, such Mg and Ag [32–35]. Although Ag addition alone does not harden the alloy to a great extent, the effect of Mg + Ag gives a synergic effect in the alloy [35,36]. In general, Mg and (Mg + Ag) addition promotes the precipitation of the $T_1$ phase at the expense of the $\theta''$ and $\theta'$ precipitates [21,28,30,32]. The typical size and distribution of the $T_1$ phase was shown to be generally finer and finely dispersed than the $\theta''$ and $\theta'$ precipitates [35–38].

On the other hand, it was shown that by Mg addition, precursor Mg-Cu phases are formed along with tangled and/or free dislocations [32]. This process actually starts very early upon aging, and further develops to GP zones and then an $S'$ phase together with a high density of $T_1$ precipitates. These two phases usually dominate the microstructure

of the T6 tempered Al-Cu-Li-Mg-X alloys. A threshold Mg content of ~0.1 at.% was found to be necessary to activate the combined precipitation of both $S'$ and $T_1$ phases [32]. The addition of Ag was found not to change significantly the precipitate sequence upon T6 tempering.

Precipitation of ordered $\delta'$ and $\delta$ precipitate phases generally occurs at the earliest stages of age hardening. These are fcc spherical or lenticular shaped precipitates, oriented to have the $(111)_\delta \mid\mid (111)_{Al}$ planes [15,16,27]. It is worth to mention that, according to Duan et al. [39], the lenticular $\delta'$ precipitates generally start forming after and grow at the expense of the $\theta'$ precipitates, thus preventing and inhibiting the side lateral growth of these latter precipitates.

The $S'/S$ phase is orthorhombic with crystallographic parameters of $a = 4.0$ Å, $b = 4.6$ Å, $c = 7.1$ Å, *Pmm2* space group symmetry, and a typical lath- or rod-like morphology. This phase has a lath habit plane $(100)_S \mid\mid (100)_{Al}$ matrix, and a crystallographic relationship of $[001]_{S'} \mid\mid [210]_{Al}$ [40]. This phase was debated to change in stoichiometric composition when it reaches the stable and coarser form of $S$, possibly being ($Al_5Cu_2Mg_2$) or ($Al_{13}Cu_7Mg_8$) ([37] and references therein).

Recently, Decreus et al. [17] studied the effect of the Cu/Li ratio on the precipitation sequence of Al-Cu-Mg-Li–X alloys. They found that at the earlier stages of aging, the microstructure was dominated by Cu-rich clusters, when Li addition is low, and by the $\delta$ phase, as Li increases in the alloy composition. Yet, the peak-aging microstructure was characterized by a large number fraction of the $T_1$ phase.

In fact, the three major phases usually formed in aged Al-Cu-Mg-Li-X alloys are $\theta$, $\delta$, and $T_1$ [39]. The $\Omega$ phase is reported to form when Ag is added to quaternary Al-Cu-Mg-X alloys, whose exact chemical composition, habit planes, and crystallographic symmetry strongly depend on the amount of the added elements and the nature of the elements added in the Al-Cu-Mg-based alloy [41–43].

On the other hand, the addition of scandium to aluminum alloys, and specifically to AA2000 series, is known to effectively pin cell and grain boundaries under plastic deformations and recrystallization processes [44–51]. In fact, scandium is generally added to promote the formation of thermally stable nanometric $Al_3Sc$ dispersoids, which effectively act as grain recrystallization inhibitors with a cell/grain boundary pinning effect. It is known that scandium addition to aluminum alloys, during solidification processes, forms fine precipitates of the type $Al_3Sc$, whose typical size is well within 100 nm [52]. These are generally called dispersoids, as they precipitate at the grain boundary. In general, following post-solidification processes, such as homogenization, hot rolling, cold rolling, and solution heat treatment, a large fraction of Al3Sc particles dissolve into the Al matrix, except for some retained fraction. For instance, solution treatment, followed by artificial aging, can favor re-precipitation of very fine $Al_3Sc$ particles. There main advantages in adding Sc to Al-alloys, especially the age-hardenable alloy series, is to strongly contribute to strengthen the alloy during casting or welding processes, and to promote the resistance to recrystallization and enhanced superplastic properties, through the presence of the fine $Al_3Sc$ dispersoids. In fact, $Al_3Sc$ exerts a significant and effective pinning action, i.e., a Zener-drag effect, chiefly at the grain boundaries. The Al3Sc dispersoids have an fcc L12-type phase structure mainly coherent with the aluminum matrix. The pinning action is responsible for an extremely high effectiveness of reducing alloy recrystallization. In particular, during all the rolling processes, also including severe plastic deformation such as Equal-Channel Angular Pressing (ECAP), the presence of the $Al_3Sc$ dispersoids affects the way in which the rolling and extrusion processes interact with the aluminum matrix.

According to the here reported complex precipitation sequences and secondary-phase precipitation variety, it is considered of primary importance the identification of the existing differences between the precipitates, and to determine their specific alloy hardening contributions.

On the other hand, it is well known that pre-strain, or even plastic deformation techniques applied before or after annealing and aging to peak hardness is able to influence

the whole secondary-phase precipitation sequences in aluminum alloys [53–58]. In this respect, severe plastic deformation techniques applied to aluminum alloys was reported to significantly modify the alloy response to T6 temper. Thus, more specifically, due to the variety of hardening phases induced by the T6 treatments in Al-Cu-Mg-Li-X alloys, severe plastic deformation (SPD) are likely to significantly change the precipitation kinetics of most, if not all, the secondary phases [14,59]. Among the different SPD techniques, the equal-channel angular pressing (ECAP) is surely one of the most important cost-effective techniques [58,60–64]. During ECAP, the material is forced to plastically deform by shearing at the intersection of the angular channels. The sample retains the same cross-sectional area after pressing, so that it is possible to repeat the pressing several times.

In this respect, previous works by one of these authors showed that the different strain paths to which tempered aluminum alloys can be subjected by ECAP are able to effectively influence the secondary-phase precipitates volume fraction [65,66]. Indeed, this did not induce a marked deterioration of other important properties such as corrosion resistance, as reported by Vicerè et al. [67,68]. In ECAP, samples are usually in form of cylindrical or square-section rod billets that are continuously forced to follow a linear path into a L-shaped equal-channel [69–78]. Strain paths in ECAP correspond to different processing routes [70,71]. Among these, here, routes A and route C were taken into consideration. Route A involves no specimen rotation between consecutive ECAP passes, while in route C, the billet is continuously rotated by 180° between passes. In addition, an evolution of the ECAP process included a further different pathway, consisting of reversing the billet path direction between consecutive passes. This is called a forward-backward ECAP (FB-ECAP) process [79–81]. In particular, a forward-backward cycling shear deformation by ECAP, or similar techniques, was shown to induce ultrafine-grained aluminum alloys in a more effective way [80,81]. This would mean that the shear deformation path in FB-ECAP is potentially able to refine the aluminum microstructure already by few passages into the die. It is, thus, expected that the recombination of tangled dislocations to form cell boundaries, and in turn, the cell boundary evolution to grain boundary, is likely to be faster in FB-ECAP with respect to conventional ECAP shear paths.

In the present work, an Sc-added Al-Cu-Li-Mg-Ag-Zr alloy was subjected to ECAP with different strain paths (i.e., routes and modes). More specifically, billets were processed up to four passes by route A, C, and FB-ECAP route A. These two routes and different processing modes introduced tangle dislocations (TDs), cell (low-angle boundaries, LABs), and grain (high-angle boundaries, HABs) at different crystallographic planes. It was, thus, possible to determine the role of the specific shear strains to which the alloy was subjected on the precipitation kinetics of the alloy hardening secondary-phase precipitates.

## 2. Experimental Details and Methods

The chemical composition of the alloy object of the present study is reported in Table 1; this is an Sc-modified Weldalite® -type alloy.

**Table 1.** Chemical composition (wt.%) of the studied Sc-modified Weldalite®-type alloy.

| wt.% | Cu | Li | Mg | Ag | Zr | Sc | Al |
|------|------|------|------|------|------|------|------|
| | 5.4 | 1.3 | 0.40 | 0.40 | 0.20 | 0.50 | bal. |

ECAP samples were prepared as cylindrical billets of 9.8 mm in diameter and 100 mm long by machining from extruded bars. The billets were then solution-treated in an air standard convection furnace at 815 K for 4 h, followed by water quenching. To determine the aging hardness peak, the alloy was aged at 460 K for duration ranging from 10 to 2880 min (10 min to 2 days).

ECAP was performed on annealed billet at the same aging temperatures of 460 K. An open die, consisting of a block of SK3 tool steel (Fe-1.1 %C) and fastened with steel bolts, was used. ECAP was carried out under uniaxial pressing forces ranged between

40–80 kN and a pressing speed of 100 mm/min. The ECAP L-shaped channel had a circular cross-section diameter of 10 mm, consisting of two linear parts intersecting at an angle $\Phi = 90°$ with a curvature extending over an angle $\Psi = 20°$. Based on this two-channels geometry, a cumulative shear strain $\varepsilon_{eq} = 1.08$ was imposed to the billet at each pass [62,70,76]. Microstructure inspections and microhardness tests were performed along the ECAP *y*-plane by cutting slices of the billet as to make measurements at the center zone of the billet. This plane is the one containing the extruded and the plastic shear directions, as depicted in Figure 1. ECAP route A does not involve any billet rotation between passes, route C consists of rotating the billet by +90° at each consecutive passage into the die, and the variant forward-backward consists of inverting the billet head at each ECAP pass during route C. Microstructure inspections were carried out after ECAP-A/4, ECAP-C/4, and FB-ECAP-A/4. Inspections were carried out both after ECAP and after ECAP followed by aging at 460 K. In this latter case, the inspections were carried out at the T6 alloy metallurgical state, and at the annealing time to reach the hardness peak after, respectively, ECAP-A/4, ECAP-C/4, and FB-ECAP-A/4.

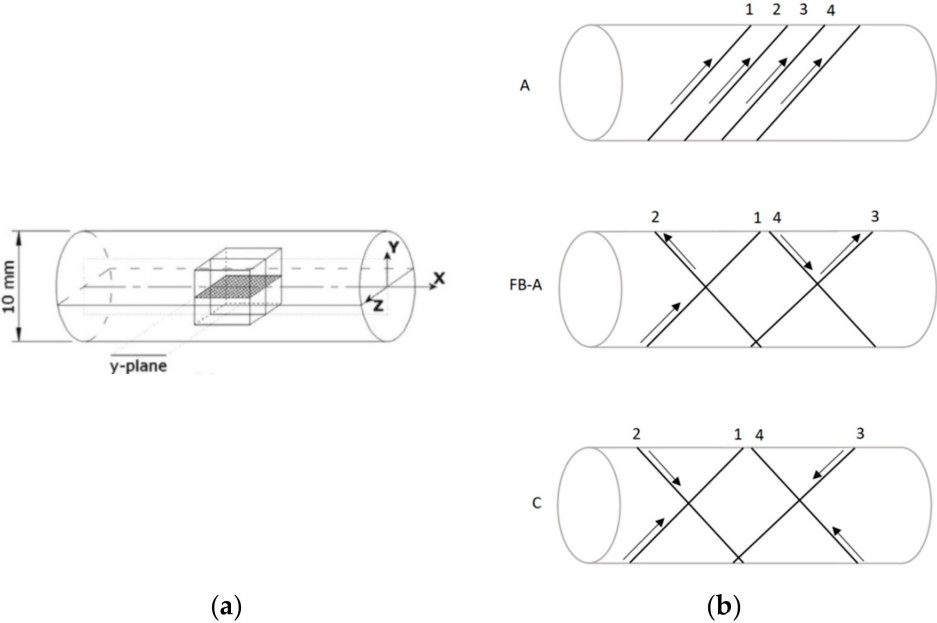

(a)         (b)

**Figure 1.** (**a**) Characteristic deformation directions in the Equal-Channel Angular Pressed (ECAP) billet. The *Y*-plane contains the shear deformation bands induced by ECAP and it is the one selected for TEM inspections. ECAP direction is along the billet *X* direction; (**b**) scheme of the activated shear planes in routes: A, FB-A, C.

Transmission electron microscopy (TEM) samples were ground and polished to ~150 µm, punched into discs of 3 mm diameter, and then electropolished by a Gatan$^{TM}$ Tenupol-5$^{®}$ (Struers, Westlake, OH, USA) working at 12 V and using a solution of 1/3 nitric acid in methanol at T = −35° C.

TEM discs were examined in a Philips$^{TM}$ CM20$^{®}$ (Philips electron Microscopy section, now FEI, Hillsboro, OR, USA) operated at 200 kV, using a double-tilt specimen holder equipped with a liquid nitrogen cooling stage. Secondary-phase precipitates were identified by using selected area electron diffraction (SAED). Thin foil thickness, *t*, was measured on TEM by diffracted beam intensity variation under dual beam conditions, using converged electron beam diffraction (CBED) patterns. This way, by plotting the linear interpolation of data points in a $S_f^2/n_{fringes}^2$ vs. $n_{fringes}^{-2}$ graph, where $S_f$ is the fringe spacing, and $n_{fringes}$ the number of counted fringes, $t_{TEM}^{-2}$ was determined at *y*-axis line intercept. The error due to the invisible dislocations (i.e., the ones oriented as to have $b·g = 0$, where $b$ is the Burgers vector and $g$ refers to the crystallographic dislocation plane) is within the experimental error of the foil thickness evaluation.

Tangled dislocation (TD) density, $\rho_{TD}$, was calculated through the count of interception points between the mesh and the existing dislocations, $n_{disl}$, in the TEM micrographs. This was evaluated by $\rho_{TD} = 2n_{dis1}/(l_{mesh}t_{TEM})$, where $l_{mesh}$ is the total length of the mesh and $t_{TEM}$ is the thickness of the TEM foil. Cell (low-angle boundary, LAB) and grain boundary (high-angle boundary, HAB) misorientation were measured by Kikuchi band patterns. The misorientation angle measurement procedure by Kikuchi pattern on TEM is fully described elsewhere in a previous published work by this author [75,76,82,83]. TEM quantitative analyses of secondary phase precipitates were carried out on crystallographic $Al_{002}$, $Al_{111}$, and $Al_{210}$ planes, depending on the habit plane of the different existing phase; tangled dislocation density was measured along the crystallographic $Al_{002}$ planes.

All the statistical evaluations were carried out according to conventional stereology methods [82] with the help of Image pro-plus® (Media Cybernetics, Inc., Rockville, MD, USA) analysis software.

Polarized optical microscopy was carried out by surface polishing and electro-chemical etching at room temperature using a solution of 4% $HBF_4$ in methanol at 20 V for few seconds.

A Remet$^{TM}$ HX-1000® (Remet S.A.S., Casalecchio di Reno, BO, ITALY) microhardness tester was used to carry out at least 12 measurements per each experimental condition. To determine the curve peak, hardness was measured on the ECAP *y*-plane surfaces after ECAP-A/4, ECAP-C/4, and FB-ECAP-A/4, and post-ECAP aging at 460 K for times ranging from 10 to 2880 min (2 days).

## 3. Results

### 3.1. T6 Temper Microstructure

The T6-hardening treatment consisted of annealing at 813 K/4 h, water quenching, and aging at 460 K for time, *t*, of 10, 20, 30, 60, and 90 min, 2, 4, 6, 8, 10, and 16 h, and 1 and 2 days. Figure 2a shows the T6 hardness, *H*, vs. aging time. It resulted a marked hardness peak at 460 K/6 h. This T6-peak condition corresponded to the formation of different nanometric precipitates, such GP zones, $T_1$-($Al_2CuLi$), $\delta'$-($Al_3Li$), and $\theta''/\theta'$-($Al_2Cu$) precipitates. Boundary pinning $\beta$-($Al_3Zr$) dispersoids also characterized the T6 alloy microstructure. These hardening secondary-phase precipitates were observed by TEM inspections and Figure 2b,c shows representative bright-filed (BF)-TEM micrographs; in particular, Figure 2b,c are TEM micrographs showing the $Al_{111}$ and $Al_{002}$ planes, respectively. On the $Al_{002}$ plane (Figure 2b), GP zones, a nanometric $\theta'$ disc, and $\delta'$ spherical precipitates were found within the grains. Grain boundaries were decorated with larger rounded $\beta'$-$Al_3(Zr,Sc)$ dispersoids. On the $Al_{210}$ plane (Figure 2d), GP zones and a few $\theta'$ disc precipitates were detected.

T6 treatment showed a significant mean grain size reduction, from the as-received $D_g = 36 \pm 3$ μm, down to $D_g = 24 \pm 2$ μm, with an aspect ratio of 0.94. Correspondingly, equiaxed grained structure was also induced to form (Figure 2e). It resulted that under annealing at 813 K/4 h, alloy grain size reduced by almost 30%. This is essentially due to the presence of the Sc- and Zr-dispersoids that effectively pinned the grain boundary coarsening tendency during annealing. In fact, it is well known that Sc and Zr form thermally stable $\beta'$-$Al_3(Sc,Zr)$ spherical dispersoids that effectively pin the grain and cell boundaries [44,45,82–86]. In particular, in [83], it was reported that Sc diffusion becomes significant from 523 K, and that the $Al_3(Sc_{1-x},Zr_x)$ dispersoid particles formation is controlled by Sc diffusion below 623 K, where Zr diffusion becomes effective. Then, at the annealing temperature of 813 K $Al_3(Sc_{1-x},Zr_x)$, dispersoids are formed and they remain stable upon further aging. The effectiveness of the grain, cell boundary, and dislocation pinning effect exerted by the nanometric size $Al_3(Sc_{1-x},Zr_x)$ dispersoids not only comes from their stability at the aging temperatures, but also originates from their crystallographic coherency with the Al-matrix. This latter aspect is documented in Figure 3, where the Ashby-Brown strain contrast characterizes the shown $\beta$-$Al_3(Sc_{1-x},Zr_x)$ dispersoids.

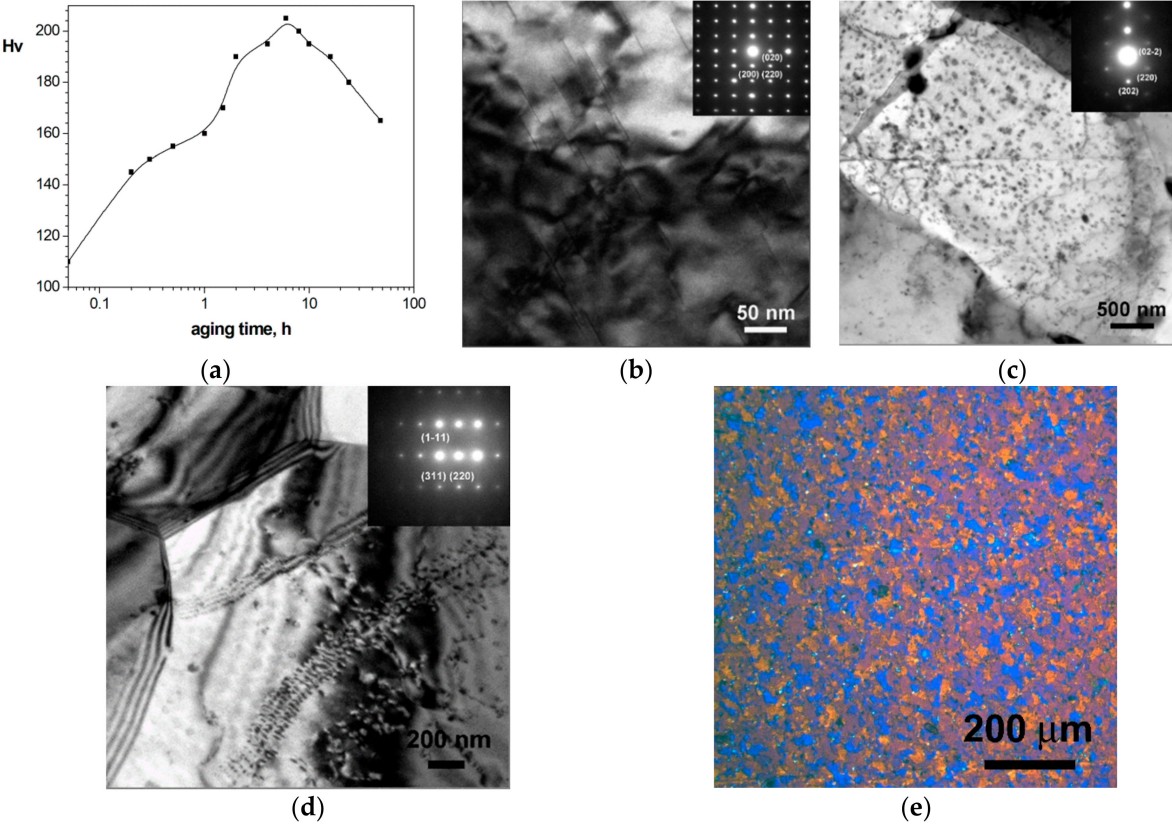

**Figure 2.** Hardness, *H*, vs. aging time, *t*, of the alloy annealing at 813 K/4 h and aging at 460 K, (**a**), BF-TEM showing the microstructure at T6 hardness peak condition (aging at 460 K/6 h), $[002]_{Al}$, (**b**), $[111]_{Al}$, (**c**), and $[210]_{Al}$, (**d**). Polarized optical micrograph showing the alloy T6 grained structure, (**e**).

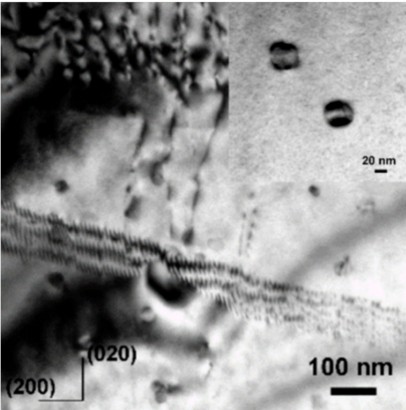

**Figure 3.** $Al_3(Sc_{1-x},Zr_x)$ dispersoids showing Ashby-Brown strain contrast in the alloy tempered at 813 K/4 h + aging at 460 K/6 h. Inset is a detail of two $\beta'$ dispersoids (dark rounded particles) showing an Ashby-Brown contrast.

### 3.2. Microstructure after ECAP

ECAP was carried out at 460 K by three different routes: route A (no rotation between passes), route C (180° rotation between passes), and forward-backward pressing (FB-ECAP) following route A. In all the three modes, the billets were pressed four times into the ECAP die (respectively, ECAP-A/4, ECAP-C/4, and FB-ECAP-A/4). The alloy was ECAP after annealing at 813 K/4 h and water quenching to room temperature. The main purpose for comparing the three SPD ECAP shearing modes was to correlate the specific shear path and deformation to the induced different Al-Cu-Li-Mg-Ag-Zr-Sc alloy secondary-phase

precipitates evolution. In fact, it is well-known that during ECAP, the alloy is subjected to a thermo-mechanical stress in terms of both shear bands that run at a typical angle of 40–42° with respect to the exit direction (*x* direction in Figure 1) [62,72,74] and adiabatic heating [76].

TEM inspections showed a significant influence of the ECAP shear deformation on the secondary-phase precipitation kinetics. In particular, ECAP was able to multiply the sites of GP zone agglomeration and secondary phase precipitation through tangled dislocations and cell boundary formation. Figure 4 reports representative TEM micrographs of the ECAP-A/4, ECAP-C/4, and FB-ECAP-A/4 conditions. The figure clearly shows that the tangled dislocations act as preferential sites of GP and secondary phase formation.

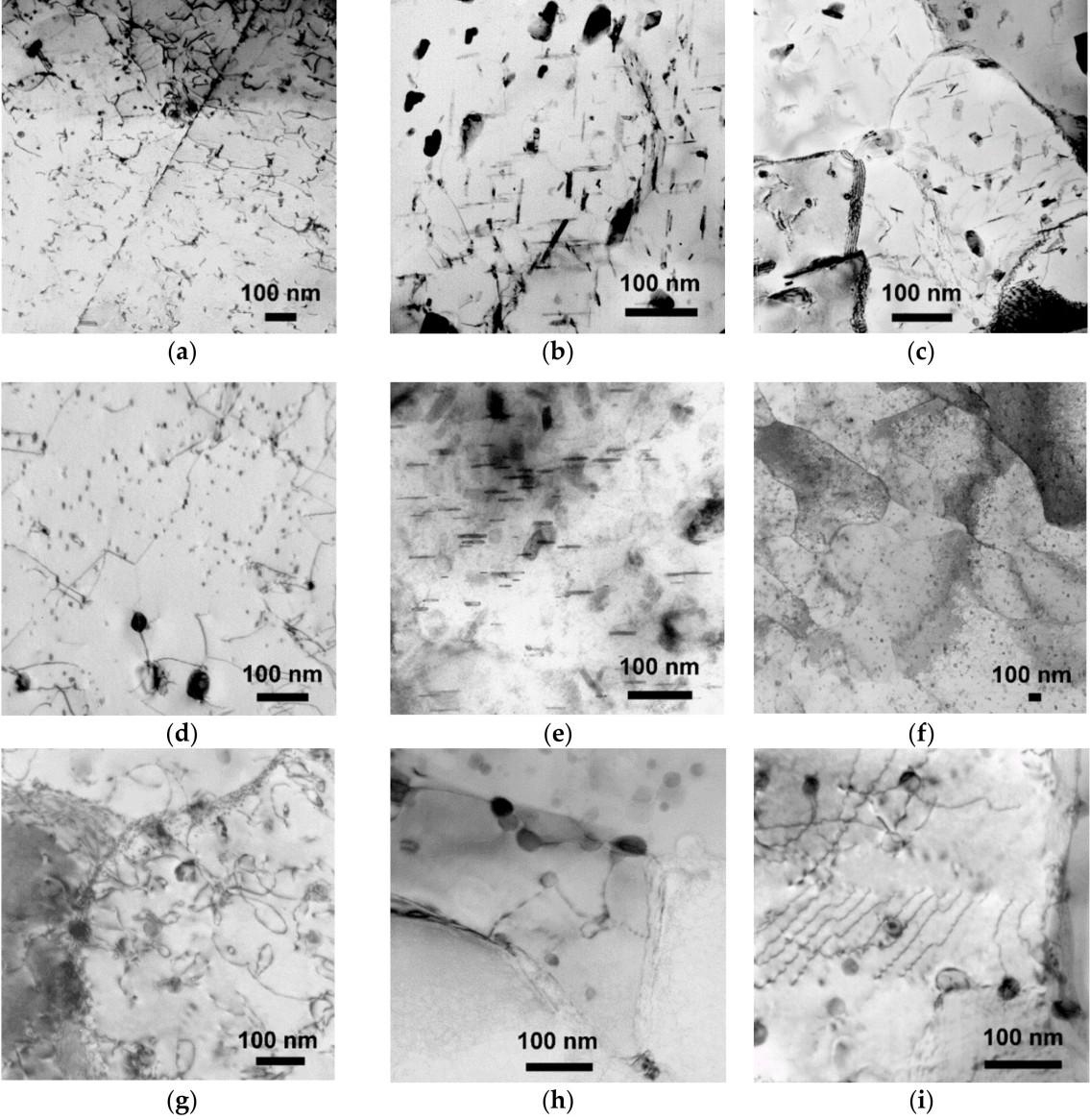

**Figure 4.** BF-TEM of ECAP-A/4, where the crystal is oriented to show $(100)_{Al}$, (**a**), $(110)_{Al}$, (**b**), and $(111)_{Al}$, (**c**); ECAP-C/4 along $(100)_{Al}$, (**d**), $(110)_{Al}$, (**e**), and $(111)_{Al}$, (**f**); FB-ECAP-A/4 along $(100)_{Al}$, (**g**), $(110)_{Al}$, (**h**), and $(111)_{Al}$, (**i**). Al-Cu clusters and GP zones were detected along the three planes; tiny needle-like $\theta$-$Al_2Cu$ phase particles were detected along $[100]_{Al}$; coarser needle-like $T_1$-$Al_2CuLi$ phase particles were detected along $[110]_{Al}$; rounded and somewhat quasi-equiaxed $\delta$-$Al_3Li$ phase particles were detected along $[111]_{Al}$.

Evaluation of the tangled dislocation density in the three experimental cases allowed correlating them with the ECAP-induced GP and secondary-phase precipitation. It showed

that the tangled dislocation density, $\rho_{disl}$, was the highest in the FB-ECAP-A/4, a little lower in the ECAP-A/4, and the lowest in ECAP-C/4 (Figure 4). That is, the following hierarchy of tangled dislocation density can be drawn: FB-ECAP-A/4 > ECAP-A/4 > ECAP-C/4. The highest dislocation density corresponded to the highest amount of GP agglomeration along the dislocation line defects and to a clear initiation of secondary-phase formation sequence. Thus, the early precipitation sequence of secondary-phase precipitation started homogeneously both at dislocation lines and within the Al grains (Figure 4). These are: $\theta$ at the Al$_{100}$ plane, $T_1$ at the Al$_{110}$ plane, and $\delta$ at the Al$_{111}$ plane. Thus, the shear deformation during ECAP at 460 K, the added adiabatic heating to which the alloy was subjected during the four passes, was responsible for the formation of GP zones and initial precipitation sequence along tangled dislocations of $\theta$, $T_1$, and $\delta$ secondary phases.

On the other hand, it is well-known that ECAP is able to significantly refine the grained structure of aluminum already at the early stages of plastic deformation. In the present case, the mean grain size, after all the three ECAP routes, was measured by a statistical meaningful number of individual measurements. These were at least 250 per condition, and the identification of grain and cell structures was carried out by means of Kikuchi band analyses. This is fully described elsewhere in previously published works by the present author ([76,82,83] and references therein). A grain was identified as a portion of the crystal all-around surrounded by high-angle boundaries (HABs). On the other hand, a cell was identified as a matrix portion surrounded by at least one low-angle boundary (LAB) line. Figure 5 reports representative low-magnification BF-TEM micrographs showing the refined grained structure after ECAP-A/4, ECAP-C/4, and FB-ECAP-A/4.

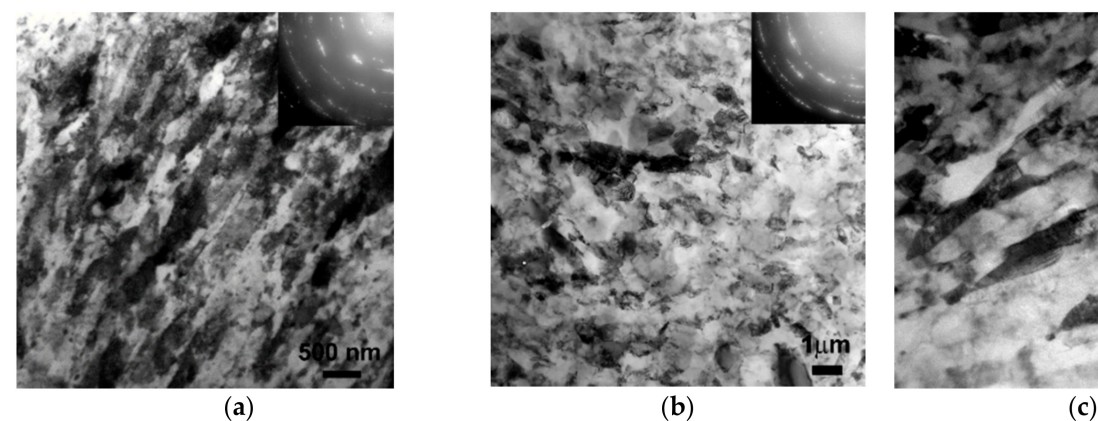
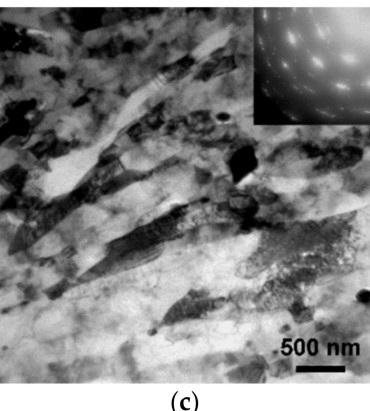

| (a) | (b) | (c) |

**Figure 5.** LM BF-TEM showing the alloy grain structure along [110]$_{Al}$ after ECAP-A/4, (**a**), ECAP-C/4, (**b**), and FB-ECAP-A/4, (**c**). Insets are Selected Area Diffraction Patterns (SAEDPs) showing the process of grain refinement through formation of concentric diffraction rings.

As reported in the histograms of Figure 6, mean grain size was significantly reduced to 1 to 3 μm, depending on the ECAP route. That is, the strongest grain size reduction was obtained after ECAP-C/4, with ECAP-A/4 being the one with the coarser refined grained structure. Thus, the following hierarchy of grain size reduction can be drawn: ECAP-C/4 > ECAP-A/4 > FB-ECAP-A/4. This means that the ECAP-C/4 resulted to be the most effective path for grain size reduction, with minimal dislocation dispersion to form tangled dislocations, rather than cell boundaries. Cell boundary density in ECAP-A/4 was lower with respect to the other ECAP route microstructures, since most of them contributed to form both cell and grain boundaries. This finding agrees well with other previously published results [56–58,60–62] and with results reported by the present author elsewhere in other aluminum alloys [75,76].

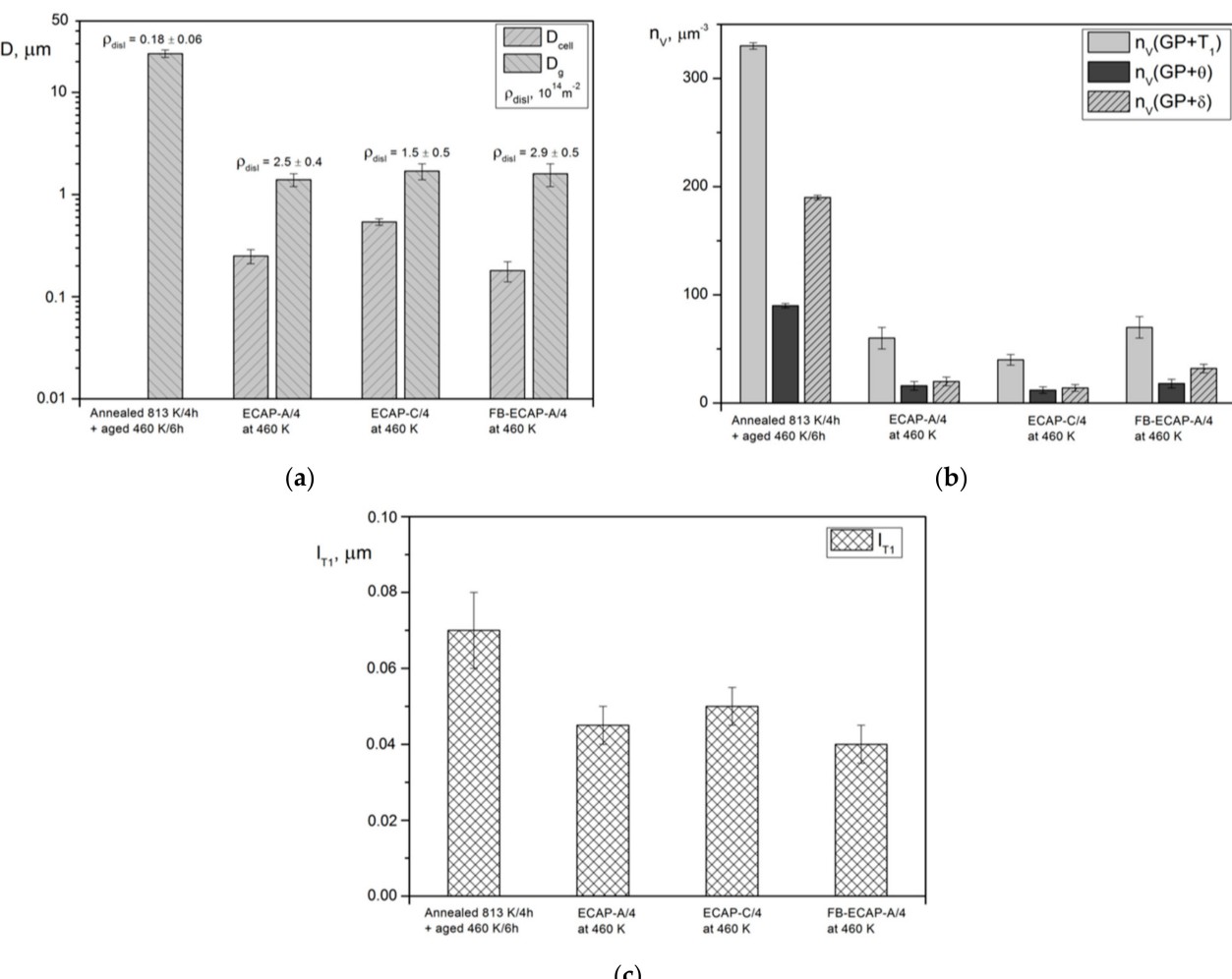

**Figure 6.** Tangled dislocation density, $\rho_{disl}$, cell, $D_{cell}$, and grain, $D_g$, size, after ECAP-A/4, ECAP-C/4, and FB-ECAP-A/4 at 460 K, (**a**); measured mean number density of GP zones and precipitates, $n_V$, of all detected secondary phases $T_1$, $\theta$, and $\delta$ induced to form under ECAP-A/4, ECAP-C/4, and FB-ECAP-A/4 at 460 K, (**b**); mean longer edge lengths of the $T_1$ platelets, $l_{T1}$, at alloy T6-condition and after ECAP at 460 K are also reported, (**c**).

The Al-Cu-Mg-Li-Ag-Zr-Sc alloy was subjected to ECAP at 460 K by the same routes and by post-ECAP aging at the same T6 conditions as obtained by the plot of Figure 2a, that is, at 460 K/6 h. Figure 7 shows the microstructure post-ECAP aging at 46 0 K/6 h, and in all the three SPD routes, a certain amount of secondary-phase coarsening with respect to the case of the undeformed T6-tempered alloy was observed (see Figure 2b–d). In particular, quite few traces of GP zones are visible. All the agglomerations and element clustering grew up to eventually precipitate as nanometric secondary phases.

Thus, the alloy microstructure was mostly characterized by the co-presence of small (few nanometric in size) to coarser (few tens of nanometer in size) secondary-phase precipitates. These were identified by SAEDPs as $T_1$-Al$_2$CuLi, $\Omega$-Al$_x$(CuLi)$_y$-type, $\theta'$-Al$_2$Cu, and $S'$-Al$_2$CuMg (Figure 8). That is, the microstructure of all the three ECAP path conditions is similar to an overaged metallurgical condition rather than a T6 hardness peak one.

Thus, these microstructure findings confirmed that both tangled dislocation and, to some extent, cell boundaries contributed to accelerate the precipitation sequences of most of the detected secondary-phase precipitates.

During post-ECAP aging, two other secondary-phase precipitates formed, $\Omega$ and $S'$. These two phases were detected to form at the same crystallographic planes of, respectively, $T_1$ and $\delta'$.

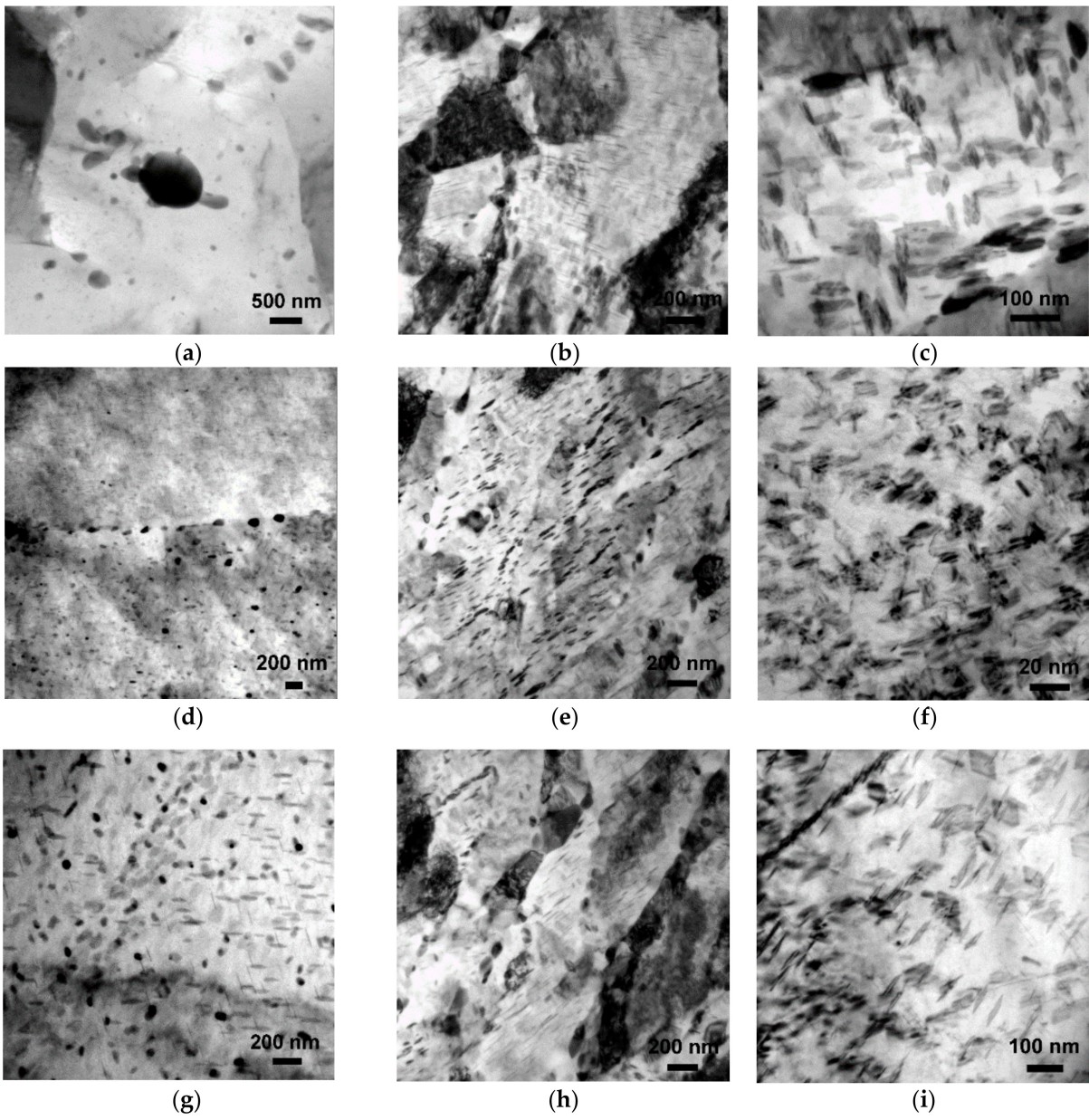

**Figure 7.** Microstructures of the post-ECAP 460 K/6 h annealed alloys, showing the different secondary-phase precipitates. BF-TEM of ECAP-A/4 along $(100)_{Al}$, (**a**), $(110)_{Al}$, (**b**), and $(111)_{Al}$ planes, (**c**); ECAP-C/4 along $(100)_{Al}$, (**d**), $(110)_{Al}$, (**e**), and $(111)_{Al}$ planes, (**f**); FB-ECAP-A/4 along $(100)_{Al}$, (**g**), $(110)_{Al}$, (**h**), and $(111)_{Al}$ planes, (**i**).

The $T_1$-Al$_2$CuLi are platelet precipitates lying on Al$_{110}$ planes with matrix coherency along Al$_{100}$ planes and $(100)_{Al}$ directions. This phase has a hexagonal crystallographic structure with *P6/mmm* space group; thus, the $[0001]_{T1}$ ǀǀ $[110]_{Al}$, and the $[1–100]_{T1}$ ǀǀ $[110]_{Al}$. This phase is considered as one of the most effective alloy strengthening in the Al-Mg-Cu-Li-X systems [87]. The Ω phase has an orthorhombic crystallographic structure ($a$ = 0.496 nm, $b$ = 0.859 nm, $c$ = 0.848 nm) with orientation relationship to the Al-matrix lattice as $(001)_\Omega$ ǀǀ $(111)_{Al}$. The $\theta$ phase has a tetragonal structure with a precipitation sequence of GP-I zone to GP-II zone ($\theta''$), $\theta'$, before reaching the final equilibrium form $\theta$-Al$_2$Cu. It lays along Al$_{100}$ planes, and it has an orientation relationship with the [001]-longer tetragonal edge ǀǀ $[100]_{Al}$. The $\delta$-Al$_3$Li precipitates have a spherical morphology fully coherent with Al$_{002}$ planes and Al$_{002}$ directions. This is a relatively thermally stable phase, as dissolution of the $\delta$ phase generally occurs at grain and cell boundaries only for temperatures higher than the ones typically used for aging [42].

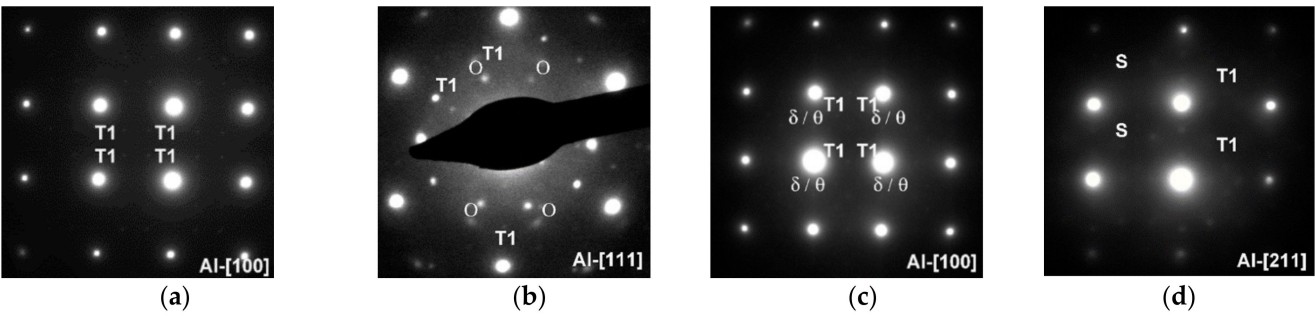

**Figure 8.** Secondary-phase precipitates identified by SAEDPs: $T_1$, (**a**), $\Omega$ and $T_1$, (**b**), $\delta$ and $\theta$, (**c**), $S$ and $T_1$, (**d**).

Figure 9 shows the statistical data of all the detected secondary-phase precipitates induced to form during ECAP at 460 K followed by aging at 460 K/6 h (T8 temper). As already documented by Figures 7 and 8, the overaged T8 ECAP + aging at 460 K/6 h conditions favored the formation of $\Omega$ and $S'$ phases. These two phases were not reported to form upon alloy T6 condition (Figure 2).

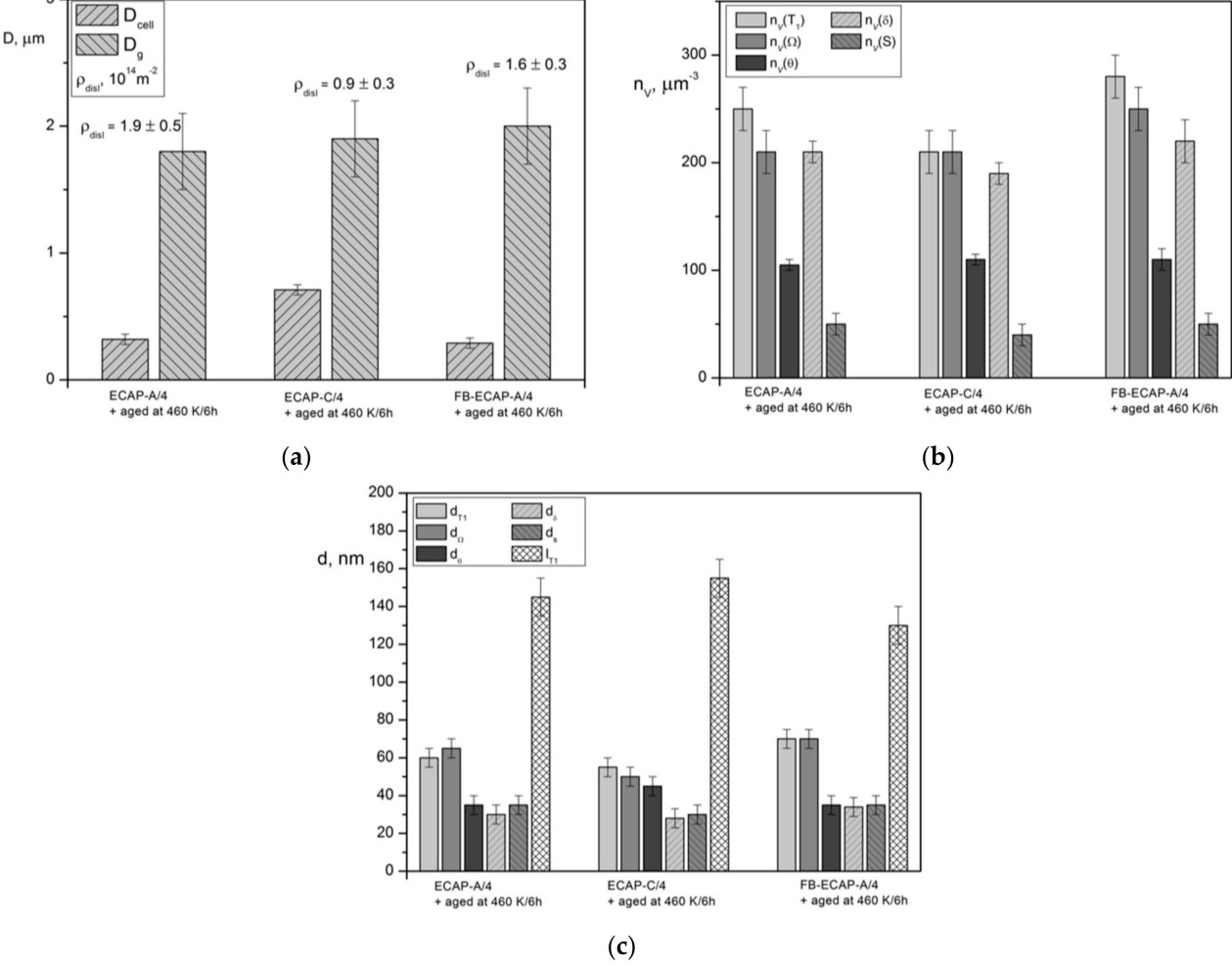

**Figure 9.** Tangled dislocation density, $\rho_{disl}$, cell, $D_{cell}$, and grain, $D_g$, size, (**a**); secondary-phase precipitate mean number density, $n_V(T_1)$, $n_V(\Omega)$, $n_V(\theta)$, $n_V(\delta)$, and $n_V(S)$, (**b**); and size, $d_{T1}$, $d_\Omega$, $d_\theta$, $d_\delta$, and $d_S$ (**c**) formed after ECAP-A/4, ECAP-C/4, and FB-ECAP-A/4 and subsequent aging at 460 K/6 h (overaged T8 condition). Mean longer edge length of the $T_1$ platelets, $l_{T1}$, is also reported, (**c**).

The figure does not include the $\beta$-$Al_3(Sc_{1-x},Zr_x)$ dispersoids, as they are thermally stable at the post-ECAP aging temperatures and they act as grain boundary-pin elements, rather than phase-hardening.

The T8 metallurgical condition induced a number density rise, $n_V(T_1)$ and $n_V(\Omega)$, from ECAP-C/4 to ECAP-A/4 and to FB-ECAP-A/4. This means that the shear path influenced the precipitation of the $T_1$ and $\Omega$ phases. The other strengthening phases did not change their number fraction with different ECAP routes. The size of all the secondary-phase precipitates was not affected by the specific ECAP shear path, as they did not change from ECAP-A/4, ECAP-C/4, and FB-ECAP-A/4.

It resulted in that the tangled dislocation density, $\rho_{disl}$, was almost double the FB-ECAP-A/4, and ECAP-A/4, with respect to the ECAP-C/4 condition. In particular, FB-ECAP-A/4 reported a tangled dislocation density quite close to the one measured in the ECAP-A/4, and thus, the shear deformation process seemed to induce a similar dislocation rearrangement during ECAP passes through these two processing routes. On the other hand, the cell and grain structure also appeared to be quite similar, contributing to confirm the microstructure evolution similarities between ECAP-A/4 and FB-ECAP-A/4. On the contrary, the lower dislocation density measured in the ECAP-C/4 is most likely due to the dislocation crystallographic recombination phenomenon induced during the 180° billet rotation between ECAP passes. Accordingly, as well-documented in hundreds of research contributions so far ([53–58,60–63,65,66,69–76,79–81,88,89], and references therein), the grain refining efficiency differences between namely route A and route C is particularly sensitive to the ECAP shear path. Thus, FB-ECAP-A/4 and ECAP-A/4 resulted to be more grain size refining-effective than ECAP-C/4.

## 4. Discussion

The microstructure findings of Figure 4 to Figure 7, and the quantitative analysis of secondary-phase precipitates (Figures 6 and 9), referring, respectively, to the ECAP and ECAP + aging at 460 K/6 h, showed a clear influence of ECAP and ECAP paths on the phase precipitation sequences. In fact, the ECAP shear deformations to which the alloy was subjected before aging to the T6 hardness peak were responsible for a significant multiplication of precipitation sites of all the secondary-phase precipitates. Indeed, it resulted that ECAP accelerated the precipitation sequence and favored the precipitation and formation of phases typically observed for longer annealing times.

In this regard, Cassada et al. [59] and Huang and Zheng [38] reported that in the presence of heterogeneous nucleation sites, such vacancies and dislocations were able to speed up the kinetics of secondary-phase precipitation in Al-Cu-Li alloys. The introduced tangled and free dislocations within grains reduced the strain energy associated with the precipitate matrix interface of the secondary phases thus promoting their nucleation and growth. The presence of Ag in the alloy particularly promoted the precipitation of the $T_1$ phase [38]. This silver effect is driven by its atomic segregation, which reduces the misfit energy between matrix and precipitate interface. Thus, the Ag nucleation and growth of $T_1$ phase is activated by the co-presence of Mg and it is promoted by a retarding of GP/$\theta'$ formation. The $T_1$ phase was detected to grow preferentially along $[100]_{Al}$ and $[111]_{Al}$ directions [90]. Limited lateral growth occurred, and the $T_1$ precipitates simply grew along the platelet longer edges (i.e., the $[0001]_{T1}$ || $[100]_{Al}$ and $[11\text{-}10]_{T1}$ || $[111]_{Al}$).

In a microstructure study of a similar Al-Li-X alloy, Jiang et al. [91] reported a $T_1$ precipitates coarsening reduced rate and even a dissolution process of the two other alloy characteristic $\theta$ and $\delta'$ phases for prolonged aging times. Anyhow, in the present case, the occurrence of these two microstructure phenomena did not occur, due to a combined effect of pre-deformation and aging that was not as drastic as a prolonged aging duration. In this sense, as reported by Shi et al. [50] with a similar Al-Cu-Mg-Li-Ag-Zr alloy, the location, number fraction, and size of the $T_1$ phase have a significant role on the mechanical properties, such as fracture toughness, of the alloy. This effect is controlled by the alloy aging conditions.

Another key microstructure aspect that was observed in the present study is nucleation and growth of a further secondary phase, quite similar to the $T_1$-$Al_2CuLi$. This is the $\Omega$ phase, which is hexagonal, similar to $T_1$ with the $Al_{210}$ habit plane, and has a chemical composition quite close to the $T_1$ phase. This phase was found to preferentially grow along $[111]_{Al}$. These findings were in good agreement with other previously published works [90,92–94]. In this respect, Garg et al. reported that the combined presence of little amount of Mg and Ag is able to stimulate the precipitation of the $\Omega$ phase in Al-Cu-Mg-Ag alloys [92], and both $T_1$ and $\Omega$ phases in Al-Cu-Li-Mg-Ag alloys [93]. Likewise, the combined additions of Mg and Ag in the present alloy are believed to be responsible for the precipitation of these phases. Ag and Mg are recognized to preferentially segregate along $[111]_{Al}$, leading to a minimum strain energy. Thus, Mg-Ag atomic co-clusters are assumed to preferably segregate along the $[111]_{Al}$, that is, on the dislocation lines and cell boundaries aligned along the $Al_{111}$ crystallographic directions. On the other hand, Cu atoms are known to preferentially segregate at grain boundaries, and thus, secondary-phase precipitation at grain boundaries is expected to be saturated by the presence of the $\theta'$-$Al_2Cu$ phase. In addition, $\Omega$ phase preferentially nucleates along $[111]_{Al}$ instead of $\theta'$, as $\Omega$ greatly reduces the matrix strain energy [92].

The combined presence of both $T_1$ and $\Omega$ is recognized as the major strengthening phases in Al-Cu-Li-X alloys, such as the Weldalite® 049-type alloys [94]. The $\Omega$ phase was reported as a phase responsible for improving the strengthening in Al-Cu-Mg-Ag and Al-Cu-Mg-Ag-X alloys [94].

As for the detected $\delta'$ precipitates and $\beta'$ dispersoids, these both have a spherical morphology. They are almost evenly distributed within the grains, and the latter is essentially present as a grain sliding pinning dispersoid, and thus, located at cell and grain boundaries. These two phases were detected to be thermally stable. That is, they were insensitive to the ECAP shearing paths, as their number fraction and size did not change significantly with different ECAP routes. As for the $\delta'$-$Al_3Li$ phase, it was found to be subjected to slight modification during shearing strain, as shearing bands were able to cut these spherical precipitates. This precipitate-cutting phenomenon is shown by the representative BF-TEM micrograph of Figure 10a. The ability of the ECAP shear band deformations to cut some of the secondary-phase precipitates was already documented by the present author in previously published works [65,66]. Other authors also reported dislocation cutting and bypassing processes during thermo-mechanical treatments of similar Al-Cu-Li-X alloys [95].

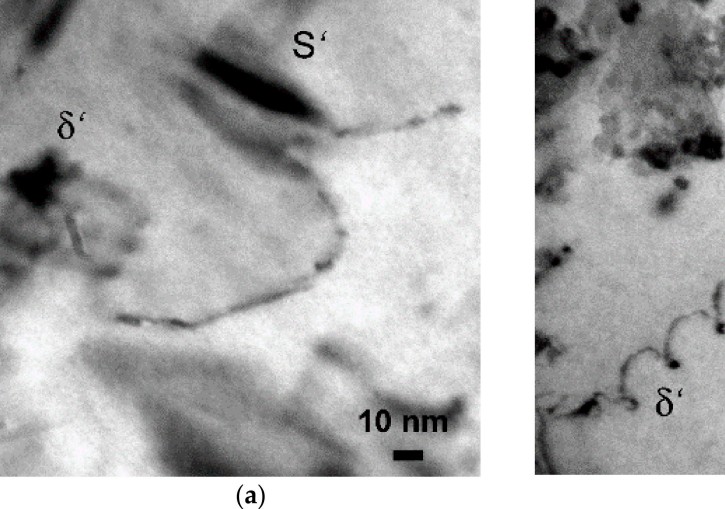

(a)            (b)

**Figure 10.** $\delta'/S'$ precipitate cutting from shearing deformation induced by ECAP, (**a**), and the helicoid dislocation pinning phenomenon occurring at $\delta'/S'$ precipitates, (**b**). TEM micrographs refer to FB-ECAP-A/4 + aging at 460 K/2 h (T8 hardness peak condition).

The *S* was slightly induced to grow by the ECAP shearing. The lath-like precipitate-growing rate was significantly lower than the one observed for the $T_1$ platelet precipitates. The *S* precipitate growth occurred along the longer lath edge, which was typically oriented along $[111]_{Al}$. This phase is able to effectively pin the sliding dislocations. The strengthening mechanism is well-documented by the dislocation helicoid-like pinning of the $S'$ nanometric precipitates (Figure 10b).

One of the $S'$ phase characteristic features is that these nanometric precipitates typically pin the sliding dislocations, thus forming helical dislocations or dislocation loops. That is, the $S'$ phase can be numbered as a strengthening phase for the present Al-Cu-Mg-Li-Ag-Zr-Sc alloy. Regarding the formation of this $S'$-$Al_2CuMg$ phase, Schneider and co-workers [90], proposed the following precipitation sequence in AA2195: GP zones—$\theta' \rightarrow T_1 + \theta' \rightarrow T_1 + \delta' \rightarrow T_1 + \theta + \delta$; for Cu/Li = 3, a precipitation sequence of GP zones + $\delta' + \beta' \rightarrow \theta'' + \theta' + \delta' + \beta' + T_1 \rightarrow T_1 + \theta + \delta + \beta$ was reported in [59]. Bai et al. [51] also reported presence of all these secondary phases in Al-Cu-Mg-Li-Ag-Zr alloy.

In the present case the precipitation sequence in the ECAP and aged Al-Cu-Mg-Li-Ag-Zr-Sc alloy was found to be as follows: GP zones + $\beta \rightarrow T_{1'} + \theta'' + \theta' + \delta' + \beta \rightarrow T_1 + \Omega + \theta + \delta + S + \beta$. That is, the following phases were here detected in the T8 ECAP + T6 aging condition and in the overaged ECAP + aged condition: $Al_2CuLi$-($T_1$), $Al_x(CuLi)_y$-type-($\Omega$), $Al_2Cu$-($\theta$), $Al_3Li$-($\delta$), and $Al_2CuMg$-($S$) precipitates, and the $Al_3(Sc_{1-x},Zr_x)$-($\beta$) dispersoids.

Micro-hardness, *H*, was measured along the *y*-plane of the three ECAP-A/4, ECAP-C/4, and FB-ECAP-A/4 conditions after ECAP and after subsequent aging at 460 K for the same times selected to determine the T6 temper hardness peak of the un-deformed alloy (Figure 2a). The plots of the *H* vs. aging time of the ECAP-A/4, ECAP-C/4, and FB-ECAP-A/4 conditions are reported in Figure 11. It appeared that the hardness peak was induced to anticipate significantly by the four ECAP passes. In particular, route A anticipated the peak to lesser extent with respect to both route C and the FB-route A.

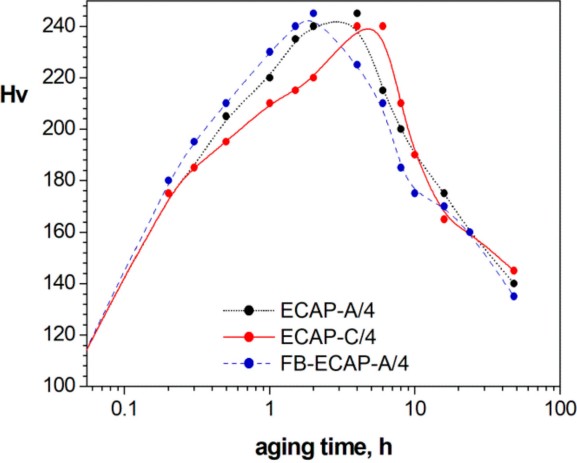

**Figure 11.** Aging curves after ECAP-A/4, ECAP-C/4, and FB-ECAP-A/4 and subsequent aging at 540 K for 10, 20, 30, 60, and 90 min, 2, 4, 6, 8, 10, and 16 h, and 1 and 2 days (same times of the undeformed alloy T6-heat treatment). Experimental errors are essentially within the datapoint.

The four ECAP passes had a similar hardness effect. It is expected that these modifications of the hardness curves with respect to the undeformed T6-temper condition are driven by a microstructure rearrangement of the grained structure and in particular by a sort of acceleration on the secondary-phase precipitation under post-ECAP aging.

Thus, based on the aging curves of Figure 11, TEM inspections were carried out at the ECAP + aging hardness peak conditions. That is, the microstructures of the ECAP-A/4 + aging at 460 K/3 h, ECAP-C/4 + aging at 460 K/5 h, and FB-ECAP-A/4 + aging at 460 K/2 h were characterized in terms of the secondary-phase precipitation (Figure 12).

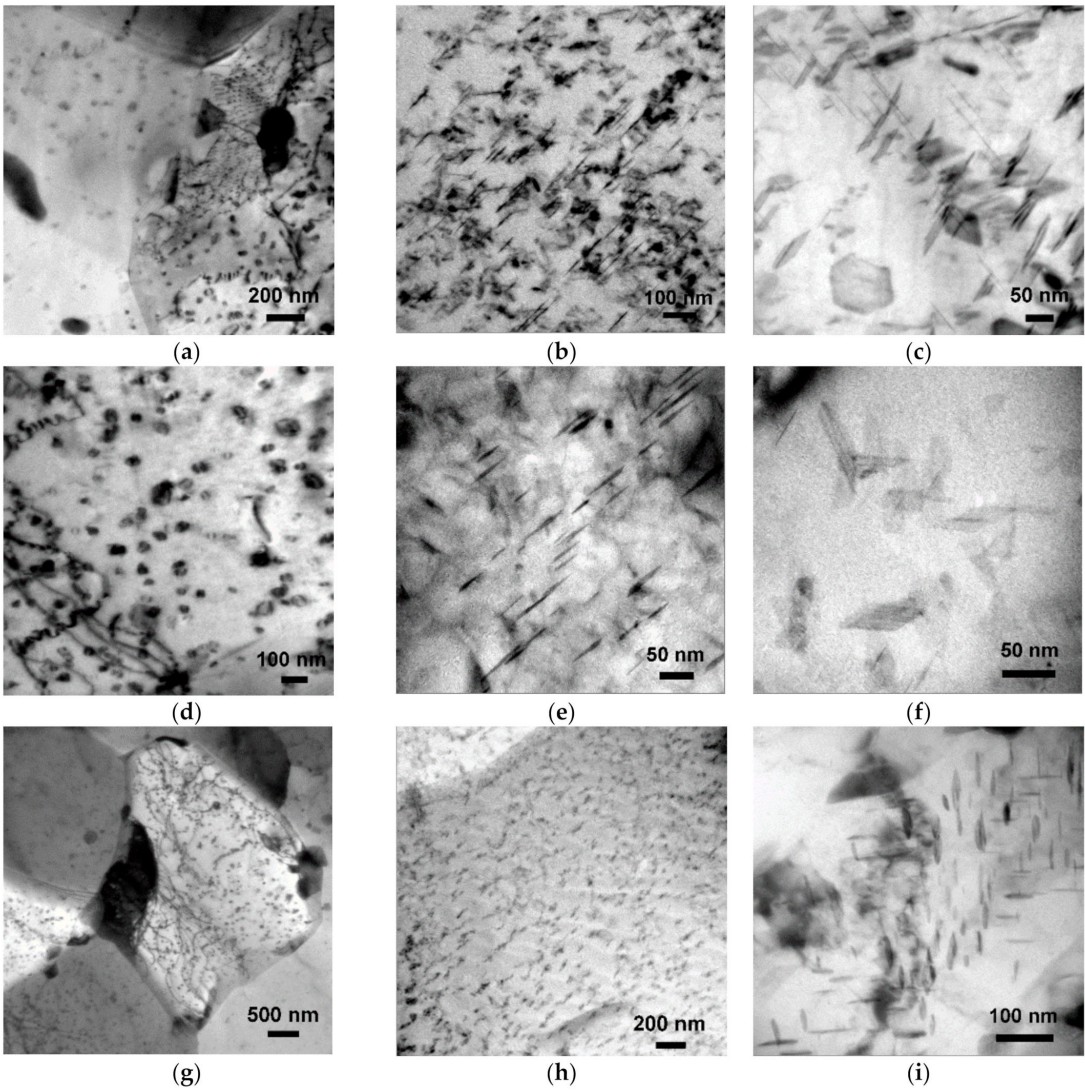

**Figure 12.** Microstructures of the post-ECAP 460 K aged alloys, with aging duration set at the maximum alloy hardness peak obtained after ECAP and subsequent aging at 460 K (3 h for ECAP-A/4, 5 h for ECAP-C/4, and 2 h for FB-ECAP-A/4). BF-TEM of ECAP-A/4 at $[002]_{Al}$, (**a**), $[111]_{Al}$, (**b**), and $[210]_{Al}$, (**c**); ECAP-C/4 at $[002]_{Al}$, (**d**), $[111]_{Al}$, (**e**), and $[210]_{Al}$, (**f**); FB-ECAP-A/4 at $[002]_{Al}$, (**g**), $[111]_{Al}$, (**h**), and $[210]_{Al}$, (**i**).

Figure 13 reports the meaningful statistics of the detected secondary-phase precipitates, and the mean cell and grain size. The microstructure of the T8 hardness peak aged condition by all the three ECAP routes here analyzed is characterized by the presence of both GP zones, secondary-phase precipitates at their early stage of formation and evolution, and some larger precipitate. The marked difference between the ECAP + aged at hardness peak (T8) and the undeformed T6 microstructures is represented by the presence of two more phases in the T8 condition. These are the $\Omega$-$Al_x(CuLi)_y$ platelet-shaped phase lying at $Al_{111}$ planes which are quite similar to $T_1$ precipitates, and $S'$-$Al_2CuMg$ lath-shaped phase lying at $Al_{210}$ planes showing a regular geometric planar morphology. The quantitative data reported in Figure 13 show little differences of the number fraction of $T_1$, $\theta'$, and $\ddot{o}'$ phases.

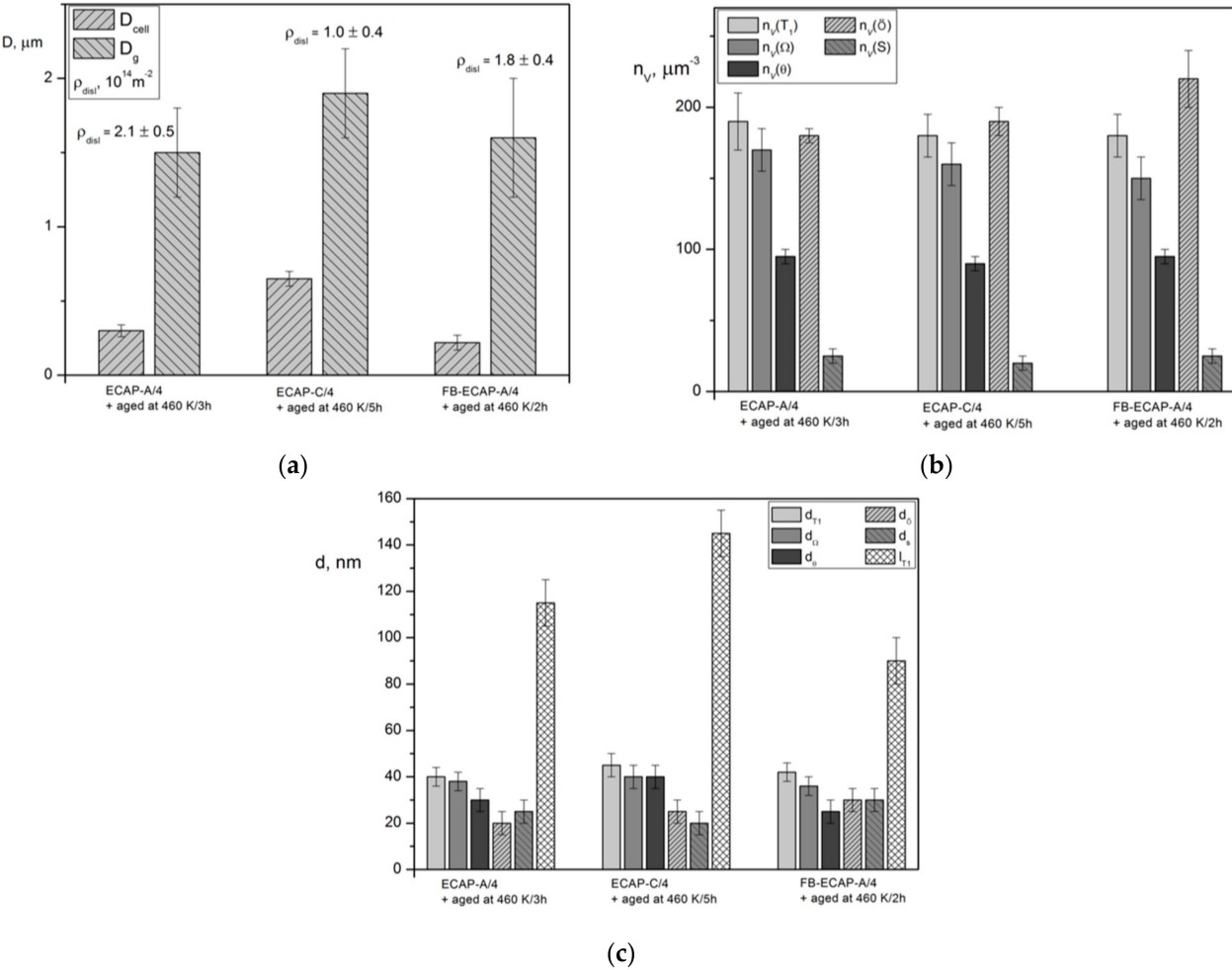

**Figure 13.** Tangled dislocation density, $\rho_{disl}$, cell, $D_{cell}$, and grain, $D_g$, size (**a**), secondary-phase precipitate mean number density, $n_V(T_1)$, $n_V(\Omega)$, $n_V(\theta)$, $n_V(\delta)$, $n_V(S)$ (**b**), and size, $d_{T1}$, $d_\Omega$, $d_\theta$, $d_\delta$, $d_S$ (**c**) formed after ECAP-A/4, ECAP-C/4, and FB-ECAP-A/4 and subsequent aging at 460 K to reach alloy hardness peak (3 h, for ECAP-A/4, 5 h, for ECAP-C/4, and 2 h, for FB-ECAP-A/4). Mean longer edge lengths of the $T_1$ platelets, $l_{T1}$, are also reported in (**c**).

On the other hand, as expected, little differences of the mean grain and cell size was found between the ECAP + aged at hardness peak, and ECAP + aged at 460 K/6 h conditions. Moreover, the little microstructure differences among the different ECAP routes (A, C, and forward-backward route A) can be considered not significant.

To better understand the secondary-phase evolution induced during T6 and the two T8 (ECAP + aging) conditions, a cumulative histogram of all the four meaningful phases, that is, $(T_1 + \Omega)$, $\theta$, $\delta$, $S$, is reported in Figure 14. Here, histogram bars, referring to all the strengthening secondary phase volume fraction, were obtained by normalizing detected phases number fraction to unity. That is, in each experimental condition reported in the *x*-axis of Figure 14, the calculated volume number, $n_V$, of each existing secondary phase was added and then normalized to 1. These were determined by TEM statistical analysis. A histogram plot like this one directly shows how each of the detected and existing secondary phase particle changes in volume number as the experimental conditions change, from undeformed, to ECAP, and then to ECAP + peak aging treatments.

It appeared that the $T_1$ phase tends to slightly reduce in favor of the $S'$ phase as the aging time increases. The two other phases, $\theta'$ and $\delta'$, did not seem to be influenced by the post-ECAP aging duration. In fact, they appeared not to change in number fraction from ECAP at 460 K to ECAP at 460 K followed by overaging at 460 K/6 h. Thus, ECAP shearing favored the formation of $S'$, while it left unaltered the kinetics of the two other

phases, $\theta'$ and $\delta'$. The kinetic evolution of the most abundant $T_1$ phase was slightly affected by the shear strain-induced formation of the $S'$ phase. Thus, the $S'$ phase only formed by the combined effect of tangled dislocations introduced by the pre-aging ECAP shear deformation, and this was not found to form upon T6-aging at 460 K/6 h.

The here reported results need to be properly discussed with previously published works and reported TEM inspections on similar alloys. Different precipitation sequences and, sometimes, different secondary-phase precipitations were reported to occur in similar Al-Cu-Li-Mg-Zr-X alloys. These findings are somehow contradictory. However, most of the Al-Cu-Li-Mg-Zr-X alloy secondary-phase precipitation findings can be summarized as follows.

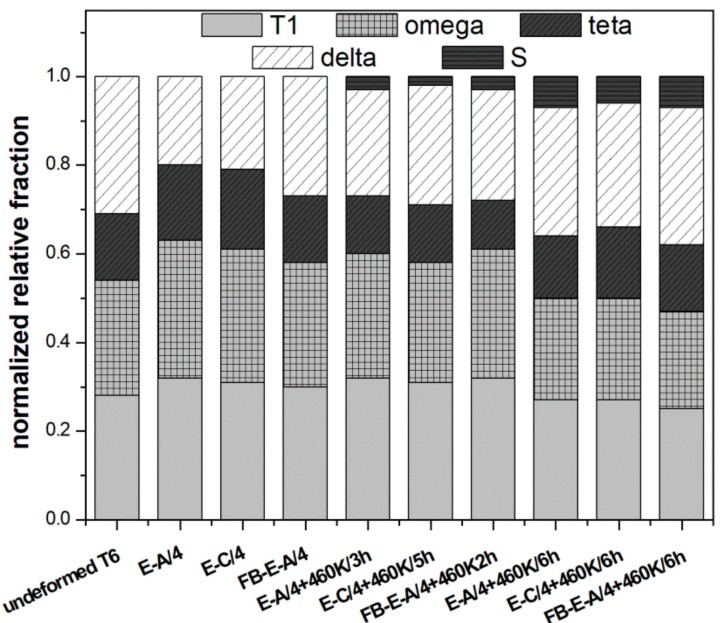

**Figure 14.** Relative frequency of the strengthening secondary-phase precipitates in T6 undeformed alloy, after ECAP-A/4, ECAP-C/4, and FB-ECAP-A/4 at 460 K, after ECAP-A/4 + aging at 460 K/3 h, ECAP-C/4 + aging at 460 K/5 h, FB-ECAP-A/4 + aging at 460 K/2 h, and after ECAP-A/4, ECAP-C/4, and FB-ECAP-A/4 + aging at 460 K/6 h. In the histogram, *E* stands for ECAP.

In some works, precipitation of the $S$-$Al_2CuMg$ was reported to occur at aging temperatures ranging from 393 to 440 K and at quite long aging times. This is the case of an Al-Cu-Mg-Li alloy where the $S$ phase was reported to precipitate at 440 K/168 h [96], or in an Al-Cu-Li-Mg-Zr-Sc alloy, where it formed at 450 K/32 h [97,98]. Moreover [27], high aging temperatures (433–450 K) of a quite similar Al-Cu-Li-Mg-Ag-Zr alloy with respect to the present alloy induced little amounts of $S$ precipitates, and only for a duration longer than 50 h. Lower aging times (20 to 60 h, at 393–450 K) to induce $S$ phase precipitation were reported for Al-Cu-Li-Mg-Zr-(Ag)-X in [13,97–99]. These aging times are still significant longer than the here reported T6-aging condition of 460 K/6 h. On the other hand, some other authors reported that by pre-aging straining Al-Cu-Li-Mg-Ag-Zr-X alloys, the formation of the $S$ phase was induced [11,100]. In these cases, the T6-aging time reduced significantly to 10–20 h. These latter findings can be considered in good agreement with the here reported results. Yet, several other TEM inspections on similar Al-Cu-Li-Mg-X alloys did not report the precipitation of $S'$ phase upon T6 temper [17,91,100–102]. In particular, no formation of an $S$ phase was reported in an Al-Cu-Li-Mg-Zr+SiC alloy aged at 435 K for up to 50 h [103], and for the duration of up to 200 h in an Al-Cu-Li-Mg-Mn-Zr [91]. No $S'$ formation was observed in another similar alloy (Al-Cu-Li-Mg-Ag-Zr, where Cu/Li = 4) for aging ranging from 373 to 450 K and times above 30 h [104]. Other research papers accounted for the preferential formation of a different phase instead of $S$, namely, $\sigma$-$Al_5Cu_6Mg_2$ in an T6-Al-Cu-Li-Zn-Mg-Mn-Zr alloy [104,105] and the $\Omega$ phase in an T6-Al-

Cu-Li-Mg-Ag alloy with a large Cu/Li ratio of 8. In this latter case, the alloy was subjected to a multi-aged T8 condition of a 6% pre-straining [105]. The post-strain aging was carried out at temperatures ranging from 410 to 455 K and duration of 12 to 60 h. Even at these extreme aging conditions of a pre-strained similar alloy, $S$ was not found. A similar but chemically different $\sigma$-Al$_5$Cu$_6$Mg$_2$ phase formation was reported instead.

The above-mentioned works refer to alloys with Cu/Li ratio typically varying from 1 to 4. Thus, the Cu/Li ratio seems not to influence the $S$ phase precipitation sequence, contrarily to what was stated in [100]. The here reported findings, according to what was reported in [11,98–100], seem to indicate that the precipitation of $S'$ phase is favored by a sequence of alloy straining and aging, or by quite long aging durations ([13,97–99] and references therein).

Moreover, Ma et al. [106], in an Al-Cu-Mg-Li-Ag-Zr-X alloy, showed a marked $T_1$ phase number fraction rise at the expense of $\theta$ phase by increasing the alloy aging pre-straining rate. Yet, two major differences can be identified in the comparison of the present findings with those reported in [106]. A first aspect consists of the different aging conditions of temperature and duration. In the present case, the alloy was subjected to both T8 and T8-overaged conditions. A second aspect refers to the significantly different pre-aging plastic deformation. In the present case, a severe plastic deformation technique was used with a cumulative strain $\varepsilon_{eq} = 4.32$, while a simple straining of up to 8% of thickness reduction was used in [106]. Finally, Ma et al. did not report the statistic evaluation of the $S'/S$ phase.

In other words, the key microstructure feature that was here identified was the ECAP + aging-driven formation of a significant amount of $S'$ phase and a small amount of $\Omega$ phase, which formed at the expense of a small amount of $T_1$. Thus, these secondary-phase precipitation modifications induced by ECAP + aging involve a lithium redistribution. In fact, the $T_1$ phase is constituted by Al$_2$CuLi, while the $S'$ phase is Al$_2$CuMg. Thus, the induced formation of $S'$ at the expense of a fraction of $T_1$ was chemically accompanied by an equally stoichiometric substitution of Li with Mg. The Li, now available in the Al-matrix, is thermally favored to form a further secondary phase. This is chemically constituted by Al + Cu + Li, and most likely as Al$_x$(CuLi)$_y$, which is the $\Omega$ phase. Based on these findings and comparison with previously published results, the ECAP + aging precipitation sequence of the present Al-3.0Cu-1.0Li-0.4Mg-0.4Ag-0.2Zr-0.5Sc alloy was GP zones + $\beta - \theta'' + \theta' + \delta' + \beta + T_1 \rightarrow \beta + \theta + \delta + T_1 + (\Omega + S - T_1)$.

The statistical data calculated for the $T_1$ phase after ECAP + aging at hardness peak and after ECAP + T6 aging (where T6 aging refers to the hardness peak of the undeformed alloy) allowed a clear identification of a precipitate-coarsening rate. The data and linear interpolation are reported in Figure 15. The figure is a plot of the mean lateral size of the $T_1$ precipitates (platelet edge $l_{T1,\Omega}$ aligned along the Al$_{111}$ directions) vs. aging time, $t^{1/3}$ for the ECAP-A/4, ECAP-C/4, and FB-ECAP-A/4 followed by hardness peak aging and by alloy T6 aging. A linear interpolation was drawn with a sufficiently sound accuracy. This plot indicated a power-law coarsening rate of the type: $l_{T1,\Omega} = K_{LSW}\ t^{1/3}$. Here, $K_{LSW} = (8/9)\ (C_{solute} V^2 \gamma D_{inter}/RT)$, where $C_{solute}$ is the phase solute concentration of Cu and Li, $V$ is the molar volume of both Cu and Li, $\gamma$ is the phase/alloy interfacial energy, $D_{inter}$ is the interdiffusion element coefficient (Cu, Li), $R$ is the Rayleigh constant, and $T$ is the temperature in Kelvin. This kind of power-law precipitate growth is in agreement with different previous models, such as the one very recently proposed by Jiang and co-workers [91]. This power-law phase growth was indeed first introduced by Lifshitz, Slyozov, and Wagner, and thus named after them as the LSW theory [107,108]. Moreover, a similar phase growth rate was also reported for $\delta$ [109], Al$_3$Sc [110], $\theta$ [111], and other thermally activated phases in Al-Cu-X and Al-Cu-Li-X alloys [112–114].

Figure 15a reports quite similar trends of $l_{T1+\Omega}$ vs. $t^{1/3}$ with little, but noticeable, rate differences among the three different ECAP shearing paths. It resulted in $l_{T1+\Omega} = 80\ t^{1/3}$ for ECAP-A/4, $l_{T1+\Omega} = 85\ t^{1/3}$ for ECAP-C/4, and $l_{T1+\Omega} = 70\ t^{1/3}$ for FB-ECAP-A/4. That is, a variation of 15 to 20% for the minimum rate was recorded in FB-ECAP-A/4, and for the maximum in ECAP-C/4. The LSW parameter $K_{LSW}$ depends, on the one hand, on (Cu

+ Li) solubility and diffusivity within the Al-matrix, and, on the other hand, on the $T_1$ to Al matrix interface energy. The first factor is essentially related to the specific chemical nature of Cu and Li with respect to Al. The second factor can, indeed, depend on crystallographic factors of the Al-matrix, such as line defects (dislocations), the activated gliding plane, and similar crystallographic factors. Thus, the small but present slope differences, that is, the $K_{LSW}$ variation, among the three different plastic deformation straining paths induced by the different ECAP routes, is likely to be due to the different tangled dislocation density and shear configuration. In fact, the FB-ECAP is the one among the three ECAP routes that induced the larger tangled dislocation entanglement. Thus, a faster $T_1$ and $\Omega$ edge coarsening occurred.

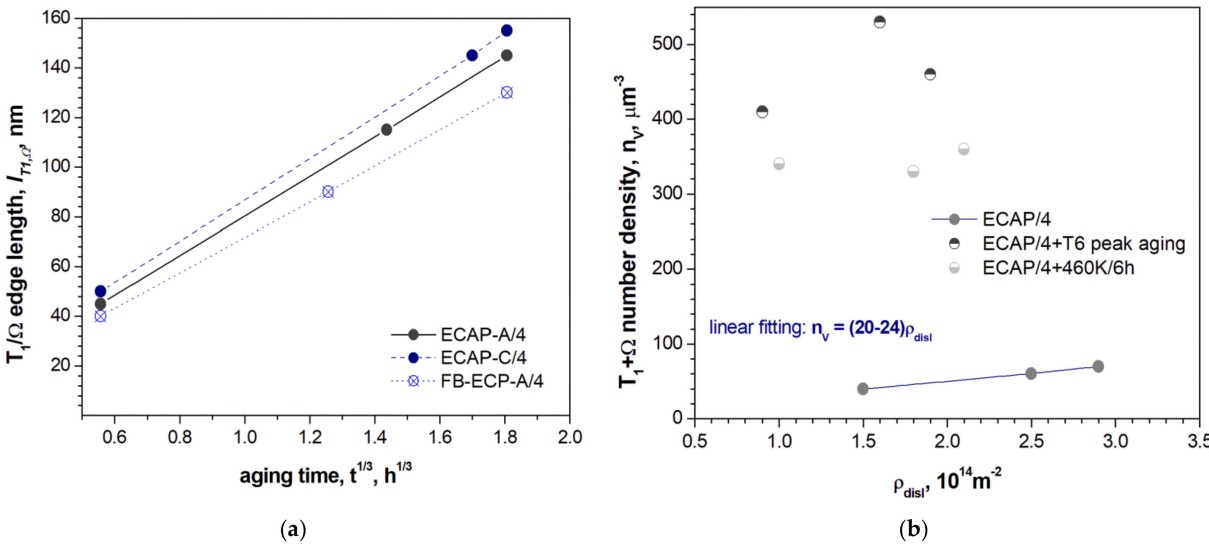

**Figure 15.** T1 + $\Omega$ precipitate power-law growth. T1 + $\Omega$ plate/lath long-edge, $l_{T1 + \Omega}$, vs. $t^{1/3}$, where *t* is the aging time, (**a**), tangled dislocation density, $\rho_{disl}$, vs. number density of the T1 + $\Omega$ precipitates, $n_V(T_1 + \Omega)$, (**b**). Data refer to ECAP-A/4, ECAP-C/4, and FB-ECAP-A/4 at 460 K, ECAP-A/4 + aging at 460 K/3 h, ECAP-C/4 + aging at 460 K/5 h, FB-ECAP-A/4 + aging at 460 K/2 h, and ECAP-A/4, ECAP-C/4, and FB-ECAP-A/4 + aging at 460 K/6 h.

As the major influence of ECAP on secondary phase precipitation sequence of existing precipitates was found for $T_1$ and $\Omega$, their ECAP-driven evolution was plotted against ECAP-formed tangled dislocations. This is because the here reported results clearly showed that both dislocation and the $T_1 + \Omega$ precipitate evolution were the most significant microstructure features affected by ECAP. Thus, Figure 15b reports plots of the number density of $T1 + \Omega$, $n_V(T_1 + \Omega)$, as a function of the tangled dislocation density, $\rho_{disl}$, where a clear relationship can be drawn. In fact, the plots confirmed a linear relationship of $n_V(T_1 + \Omega) = (22 \pm 2) \times \rho_{disl}$ for ECAP at 460 K, ECAP + aging at hardness peak, and ECAP + aging at 460 K/6 h. Thus, the choice of the ECAP path had a significant influence on the evolution of the predominant hardening secondary phases ($T_1$, and $T_1 + \Omega$) promoted by the following aging treatment. Similar results were reported by Zhang et al. [102] for $\delta'$ precipitates in an Al-Cu-Mg-Li-X alloy.

## 5. Conclusions

The secondary-phase precipitation sequences of an Al-Cu-Mg-Li-Ag-Zr-Sc alloy subjected to different Equal-channel angular pressing (ECAP) routes (A, C, and forward-backward A) followed by aging was studied. The TEM inspections and the quantitative analyses of the existing phases allowed to identify the role of the ECAP shearing and post-ECAP aging on the relative fraction of the hardening phases. Thus, the following conclusions can be drawn:

(i)    ECAP shearing and the introduced tangled dislocations were identified as responsible for a significant secondary-phase precipitation sequence acceleration. In particular, two phases were found to be greatly influenced by the presence of the tangled dislocations within the grained structure: the platelet $T_1$-$Al_2CuLi$ and the lath $S$-$Al_2CuMg$.

(ii)    ECAP + aging was responsible for the lath-shaped $S$-$Al_2CuMg$ phase precipitation, lying parallel to the $Al_{210}$ planes. This phase evolved and increased in number fraction with aging time at the expense of the $T_1$-$Al_2CuLi$ and the platelet-shaped $\Omega$-$Al_x(CuLi)_y$ precipitate phases both lying at $Al_{111}$ planes;

(iii)    ECAP + aging induced the following precipitation sequence: GP zones $+ \beta \rightarrow \theta'' + \theta' + \delta' + \beta + T_1 \rightarrow \beta + \theta + \delta + T_1 + (\Omega + S - T_1)$.

(iv)    The $T1 + \Omega$ precipitates coarsening rate with aging time was found to be of a Lifshitz-Slyozov-Wagner (LSW) type, that is, as a power-law: $l_{T1 + \Omega} = K_{LSW}\, t^{1/3}$, with $l_{T1 + \Omega}$ being the platelet and lath longer edge.

(v)    $T_1$ and $\Omega$ precipitates evolved by following a direct correlation to the tangled dislocation density introduced by pre-aging ECAP. The ECAP routes influenced both the volume fraction of the $T_1$ and $\Omega$ precipitate and their growing rate.

**Author Contributions:** Conceptualization, M.C.; methodology, M.C. and C.P.; formal analysis, C.P.; investigation, M.C. and C.P.; resources, M.C.; data curation, C.P.; writing—original draft preparation, M.C.; writing—review and editing, M.C. and C.P. All authors have read and agreed to the published version of the manuscript.

**Funding:** This research received no external funding.

**Institutional Review Board Statement:** Not applicable.

**Informed Consent Statement:** Not applicable.

**Data Availability Statement:** The data presented in this study are available on request from the corresponding author.

**Acknowledgments:** The EU-funded COST Action CA15102—Solutions for Critical Raw Materials Under Extreme Conditions (CRM-EXTREME), website: https://www.cost.eu/actions/CA15102, is acknowledged for the material supply.

**Conflicts of Interest:** The authors declare no conflict of interest.

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
