# Peer review of "High-Temperature Equal-Channel Angular Pressing of a T6-Al-Cu-Li-Mg-Ag-Zr-Sc Alloy"

_jmmp, doi:10.3390/jmmp5010006_

Round 1
Reviewer 1 Report
Congratulation for this thorough and very detailed microstructural analysis.
Thank you for the deep insight to the role of different precipitates in this alloy.
There are only some smaller minor points that should be corrected, directly marked in the paper.

Author Response
Yellow marked comments were addressed in the Revised (rev1) version of the manuscript.
Reviewer 2 Report
Authors present quite comprechensive study of Al-3.0Cu-1.0Li-0.4Mg-0.4Ag-0.2Zr-0.5Sc alloy. Study gives quite many experimental details and can be used for further development of the field.
After careful reading, I think that introduction is very long and might be shortened to have better focus. Sc-containing alloys know to have so-colled Sc-effect and usually form Al3Sc particles even if Sc content is very low. I suggest to discuss such effect in the context of current study. Authors can refere to reviews:
https://www.tandfonline.com/doi/abs/10.1179/174328005X14311
https://www.tandfonline.com/doi/full/10.1080/09506608.2015.1137692
Authors show quite comprehensive study of complex Al-based Al-Cu-Li-Mg-Ag-Zr alloy with addition Sc. Authors give quite long introduction with analysis of addition of various metals to Al alloys. Nevertheless, authors do not give any analysis of Sc addition to mentioned systems. As soon as addition of Sc was mentioned as the main point of investigation, I suggest to give more analysis of Sc effect but make shorter introductory part to make it more narrow. Sc effect is well-known for Al alloys and usually plays important and dominant role in alloy's microstructure. There are a number of reviews which address the mentioned phenomenon.
In the results part, existence of Sc-based precipitates were mentioned, but information about other intermetallic compounds given in the introduction is not so important for the story, as I can see.
In experimental part, I suggest to give direct analytical composition of alloy. Figure 11 seems to be important for the discussion. Nevertheless, it is not really clear how Figure 11 was plotted and based on what type of data, how fractions were obtained?
As general suggestion, manuscript is long and represents large amount of experimental data important for the progress of the field. It might be interesting for broader community. As general suggestion, I would recommend authors to give not only raw images and text, but try to represent all analysis as schematic diagrams and schematic plots to give more clear view for readers. It might help readers to get important information given.
I think, manuscript is well written and might be interesting for broad community. After addressing mentioned issues, manuscript can be considered further for publication.
Author Response
Please see attached word file where replies to the Reviewers' #2 comments are included.

Reviewer 3 Report
referee report
jmmp-1050027-peer-review-v1
Marcello Cabibbo and Chiara Paoletti
High-temperature Equal-Channel Angular Pressing of a T6-Al-Cu-Li-Mg-Ag-Zr-Sc alloy
The present manuscript reports on equal-channel angular pressing of a T6-Al-Cu-Li-Mg-Ag-Zr-Sc alloy and its microstructural
characterization. The manuscript is well organized, and comprises 12 figures and 115 references. The figures are well
prepared, including informative figure captions.
The authors present a long introduction, which gives a complete overview over the field. This is also manifested by the amout
of references, which give the paper a review character. The methods and procedures are well described.
Some general remarks:
# There should always be a space between the physical quantity and its unit.
# Please use SI units throughout the manuscript.
# Please insure that all abbreviations are defined at their first appearance in the text. This is especially important
concerning the review character -- not all readers are specialists in this field.
Specific comments:
# The abstract mentions ECAP route A, C, and forward-backward route A. Is it possible that a reader can understand this before
reading the paper?
Some minor corrections to the English are required as well.
So, in summary, this manuscript may be published with minor amandements.
Author Response
Please refer to the Response letter here attached.
